# *Diplodia seriata* Biocontrol Is Altered via Temperature and the Control of Bacteria

**DOI:** 10.3390/microorganisms12020350

**Published:** 2024-02-08

**Authors:** Alejandra Larach, Paulina Vega-Celedón, Diyanira Castillo-Novales, Lorena Tapia, Italo Cuneo, Fabiola Cádiz, Michael Seeger, Ximena Besoain

**Affiliations:** 1Escuela de Agronomía, Facultad de Ciencias Agronómicas y de los Alimentos, Pontificia Universidad Católica de Valparaíso, San Francisco s/n La Palma, Quillota 2260000, Chile; pvegaceledon@gmail.com (P.V.-C.); diyaniracastillonovales@gmail.com (D.C.-N.); lorena.tapia.bio@gmail.com (L.T.); italo.cuneo@pucv.cl (I.C.); fabiola.cadiz@pucv.cl (F.C.); 2Laboratorio de Microbiología Molecular y Biotecnología Ambiental, Departamento de Química & Centro de Biotecnología Dr. Daniel Alkalay Lowitt, Universidad Técnica Federico Santa María, Avenida España 1680, Valparaíso 2390123, Chile; michael.seeger@usm.cl; 3Millennium Nucleus BioGEM, Valparaíso 2390123, Chile

**Keywords:** biocontrol, Botryosphaeria dieback, *Botryosphariaceae*, demelanizing activity, *Diplodia seriata*, grapevine trunk diseases, GTDs, in vivo, in vitro, Pseudomonas, *Rhodococcus*

## Abstract

Grapevine trunk diseases (GTDs) attack the vine’s wood, devastating vineyards worldwide. Chile is the world’s fourth-largest wine exporter, and Cabernet Sauvignon is one of the most economically important red wine varieties. Botryosphaeria dieback is an important GTD, and *Diplodia seriata* is one of the main pathogenic species. Biocontrol studies of these pathogens are commonly carried out at different incubation times but at a single temperature. This study aimed to evaluate the biocontrol effect of Chilean PGPB and grapevine endophytic bacteria against *D. seriata* at different temperatures. We analyzed the biocontrol effect of *Pseudomonas* sp. GcR15a, *Pseudomonas* sp. AMCR2b and *Rhodococcus* sp. PU4, with three *D. seriata* isolates (PUCV 2120, PUCV 2142 and PUCV 2183) at 8, 22 and 35 °C. Two dual-culture antagonism methods (agar plug diffusion and double plate) were used to evaluate the in vitro effect, and an in vivo test was performed with Cabernet Sauvignon cuttings. In vitro, the greatest inhibitions were obtained using the agar plug diffusion method and at a temperature of 8 °C, where *Rhodococcus* sp. PU4 obtains a 65% control (average) and *Pseudomonas* sp. GcR15a a 57% average. At 22 °C, only strains of *Pseudomonas* sp. show control. At 35 °C, one *Pseudomonas* strain shows the highest control (38%), on average, similar to tebuconazole (33%), and then *Rhodococcus* sp. (30%). In vivo, a biocontrol effect is observed against two *D. seriata* isolates, while the PUCV 2142 proves to be more resistant to control. The biocontrol ability at low temperatures is promising for effective control in the field, where infections occur primarily in winter.

## 1. Introduction

The cultivation of grapevines (*Vitis vinifera* L.) is an important and widespread agricultural activity worldwide that dates back more than 7000 years. Grapevines are highly appreciated for their fresh fruits and wines [1]. However, one of the most significant challenges in growing grapevines is their susceptibility to various pathogens, including viruses, bacteria, fungi and nematodes [2]. Grapevine trunk diseases (GTDs) attack grapevine wood and devastate vineyards worldwide. Although GTDs have been known for more than a century, their impact has increased significantly in recent decades [3,4,5,6], causing significant production losses [4,7]. The three main GTDs are Botryosphaeria dieback, Eutypa dieback and Esca disease, which generally attack perennial organs at all stages of vine growth [2,3,8]. Botryosphaeria dieback is one of the most important GTDs worldwide and has been associated with 26 botryosphaeriaceous taxa [9]; the most frequently reported species are *Diplodia seriata*, *D. mutila*, *Lasiodiplodia theobromae* and *Neofusicoccum parvum* [2,4,9,10].

Chile is currently the top wine exporter in America and the fourth wine exporter worldwide, surpassed only by France, Spain and Italy [11]. The total area of Chilean vineyards currently occupies more than 141 thousand hectares, which has a wine production potential of close to 1200 million liters [11]. According to the National Viticultural Registry of Chile 2020, Cabernet Sauvignon is at the leading position among the red wine varieties, with more than 40,000 hectares [12]. A study of Botryosphaeria dieback disease in Cabernet Sauvignon vineyards in Central Chile showed that the overall disease incidences were 87% in 2010 and 84% in 2018, the severities of the damage were 49% in 2010 and 47% in 2018 and yield losses were 39% in 2010 and 46% in 2018, wherein *D. seriata* was the most prevalent fungus isolated from symptomatic plants [4]. Other studies demonstrated that *D. seriata* (anamorph of *Botryosphaeria obtusa*) is one of the most frequently isolated species from diseased grapevines in Chile [4,13,14,15] and other regions, such as California [16], Mexico [17], Spain [18,19], Portugal [20], France [21], Iran [22], Lebanon [23], Algeria [24], Tunisia [25] and Australia [26].

Compared to other species of *Botryosphariaceae* (*Botryosphaeria dothidea*, *Diplodia corticola*, *D. mutila*, *Dothiorella iberica*, *L. theobromae*, *N. parvum* and *Spencermartinsia viticola*), conidia of *D. seriata* showed the highest germination under a wide range of temperatures (10 to 40 °C) [27]. More isolates of *D. seriata* than *N. parvum* were isolated from colder places in Iran [22]. *Diplodia seriata* strains were isolated at temperatures between 5 and 40 °C. These studies suggest that *D. seriata* is probably the most cosmopolitan botryosphaeriaceous fungus infecting grapevines [27]. Climate change, including extreme heat or cold, will continue to increase stress on plant communities, and some species of *Botryosphariaceae*, such as *D. seriata*, will probably cause large-scale damage [28] due to their wide growth temperature range. Considering this, it is essential to know, on the one hand, how the Chilean strains of *D. seriata* behave at different temperatures and their possible biocontrol agents. During the pruning season, there is dispersion of spores, which coincides with rainfall [15], and the average temperatures oscillate between 8 and 22 °C in the Maule region of Central Chile. During the summer, the maximum average temperatures on some days exceed 35 °C (https://agrometeorología.cl, 2023 accessed on 14 August 2023) [29].

Several species of *Botryosphariaceae* are endophytes or have an endophytic stage, including *D. seriata* [28]. Currently, there are no efficient tools to eradicate infections caused by these pathogens other than surgical removal of the infected organs, so they are managed mainly by practices that aim to prevent infections [9,30]. In recent years, synthetic products for controlling Botryosphaeria dieback in *Vitis vinifera*, such as benzimidazoles (benomyl and carbendazim), have been limited and banned in several countries [30]. Biocontrol of GTDs using microorganisms is a promising alternative [31,32] that appears to be a response to the increase in wood diseases, product restrictions and low efficacy of some chemicals [30]. Bacteria from the phylum *Actinomycetota* [33,34] and the genus *Pseudomonas* [31] have shown high antifungal activity against fungi associated with GTDs.

Biocontrol in vitro and in vivo studies of grapevine trunk pathogens are commonly performed at different incubation times but only at a single temperature, usually 25 °C [31,32,33,34]. In vitro and in vivo studies have shown that the fungus *D. seriata* is able to grow at a wide range of temperatures [22,27]. Therefore, its biocontrol at different temperatures should be studied, considering those that occur in winter and summer during vineyard cultivation. The objective of this study was to evaluate the biocontrol effects of two plant growth-promoting bacteria (PGPB) bacteria isolates from wild plants from Chile and one endophyte bacteria from vine against different isolates of *D. seriata* at low (8 °C), medium (22 °C) and high temperatures (35 °C). The capacity of biocontrol was evaluated in vitro using two forms of the dual-culture antagonism method and an in vivo assay with Cabernet Sauvignon cuttings. We analyzed the biocontrol effects of plant growth-promoting bacteria (PGPB), *Pseudomonas* sp. GcR15a, *Pseudomonas* sp. AMCR2b and the endophyte *Rhodococcus* sp. PU4, on three *D. seriata* isolates at 8, 22 and 35 °C.

## 2. Materials and Methods

### 2.1. Bacteria

The psychrotolerant PGPBs for biocontrol studies were *Pseudomonas* sp. GcR15a [35], *Pseudomonas* sp. AMCR2b (protected use under patent request: PCT/CL2022/050057) and the endophyte *Rhodococcus* sp. PU4, which was obtained from a healthy and nongrafted Cabernet Sauvignon plant. Three phytopathogenic *D. seriata* isolates were used in this study: PUCV 2120, PUCV 2142 and PUCV 2183. These were obtained from diseased grapevine plants [4]. These microorganisms were obtained from the culture collections of the Escuela de Agronomía (Pontificia Universidad Católica de Valparaíso, Quillota, Chile) and the Molecular Microbiology and Environmental Biotechnology Laboratory (Universidad Técnica Federico Santa María, Valparaíso, Chile).

### 2.2. In Vitro Biocontrol Assay (Agar Plug Diffusion Method)

The biocontrol activity on *D. seriata* isolates was evaluated using the agar plug diffusion method, as described by Balouiri et al. [36], with modifications. Yeast malt (YM) agar culture medium (10 g L^−1^ glucose (Merck, Darmstadt, Germany), 3 g L^−1^ malt extract, 5 g L^−1^ peptone (Difco, Laboratories, Franklin Lakes, NJ, USA), 3 g L^−1^ yeast extract (Difco) and 15 g L^−1^ agar (Difco)) was used for biocontrol bacteria (strains GcR15a, AMCR2b and PU4), and Potato Dextrose Agar (PDA, HiMedia, Laboratories West Chester, PA, USA) was used for *D. seriata* isolates (PUCV 2120, PUCV 2142 and PUCV 2183). Bacteria were grown in YM medium for ~12 h and then adjusted to a turbidity at 600 nm of 1. Three drops of 100 µL of each isolate were placed on a plate, allowed to dry and incubated for 72 h at room temperature (22 ± 3 °C). *Diplodia seriata* isolates were grown in PDA medium for 4–5 days at 26 °C. An 8 mm mycelial disc of colony under active growth of each *D. seriata* isolate was obtained with a sterilized metal punch and deposited in a PDA plate at 4 cm of an 8 mm piece of YM agar, with bacterial growth also obtained from a colony in active growth. A negative control (YM agar medium without bacteria) and a positive inhibition control (tebuconazole, 0.3 mg L^−1^) [37] were incorporated. This assay was performed twice in triplicate. Inhibitions were observed at 7 and 14 days at 8, 22 and 35 °C. The internal radius of fungal growth was recorded, measuring from the center of the inoculum plug to the edge of hyphal growth, at 7 and 14 days of incubation.

The inhibition percentage of each treatment was calculated using the Equation (1):Percentage of inhibition (%) = ((R − r)/R) ∗ 100,(1)
where *R* corresponds to the average radius of fungal growth of the negative control (C−) and *r* to the average radius of fungal growth with the applied treatment [38].

### 2.3. In Vitro Biocontrol Assay (Double Plate Method)

The biocontrol activity on *D. seriata* isolates was analyzed using the double plate method (two-sealed-base plates), as reported by Dennis et al. [39] and Delgado et al. [40], with modifications according to Gao et al. [41]. YM agar culture medium was used for biocontrol bacteria (strains GcR15a, AMCR2b and PU4), and PDA was used for *D. seriata* isolates (PUCV 2120, PUCV 2142 and PUCV 2183). The biocontrol bacteria were streaked onto YM plates, 8 mm pieces of *D. seriata* isolates were placed in the center of PDA plates, and the two plates were sealed with parafilm. This assay was performed twice in triplicate. Fungal growth inhibition was observed after 7 and 14 days at 8, 22 and 35 °C. The fungal growth diameter was measured. The inhibition percentage of each treatment was calculated using Equation (1).

### 2.4. In Vivo Biocontrol Assay

The in vivo biocontrol capacity of the grapevine endophytic bacterium *Rhodococcus* sp. PU4 and the PGPBs *Pseudomonas* sp. GcR15a and *Pseudomonas* sp. AMCR2b was determined with three *D. seriata* isolates (PUCV 2120, PUCV 2142 and PUCV 2183), employing cuttings following the methodology of Haidar et al. [42] and Kotze et al. [43] (Appendix A). From a vineyard cv. Cabernet Sauvignon, 18 cm long one-year-old semi-lignified cuttings were taken, disinfected with 1% sodium hypochlorite for 5 min and 95% ethanol for 30 s and washed three times with sterile-distilled water (SDW). Finally, they were dried at room temperature inside a laminar flow chamber. The cuttings were kept at 5 °C for two weeks before use. A fresh wound was made and was immediately inoculated with 40 µL of each bacterial suspension (1 × 10^8^ CFU mL^–1^) in YM medium, YM medium (negative control, C−) and Tebuconazole (0.5% p v^–1^, positive control, C+). Bacterial suspensions were prepared as described previously in Vega-Celedón et al. [35]. After inoculation with the treatments, the cuttings were left at room temperature until the total absorption of the inoculum. Then, each cutting was placed in a humid chamber with a disinfected plastic grid including a sterile absorbent moistened paper (5 mL of SDW) and placed at 8, 22 °C and 35 °C. One day later, the cuttings were inoculated with a 50 µL suspension of *D. seriata* at a concentration of 1 × 10^5^ conidia µL^−1^ at the same end where the previous inoculation was performed. Fungal suspensions were prepared as described previously in Larach et al. [6,44]. Cuttings inoculated with each *D. seriata* isolate and their treatments (GcR15a, AMCR2b, PU4, C− and C+) were placed in humid chambers at 8, 22 and 35 °C. Six cuttings were used for each treatment, isolate and temperature, a total of 270 cuttings. The assay was evaluated two months after inoculation with *D. seriata*. The length of the vascular lesions was measured in every detached cutting. The inhibition percentage of each treatment was calculated using Equation (1). 

### 2.5. Statistical Analysis

One-way ANOVA was used to analyze the effects of the biocontrol strains. After performing one-way ANOVA, Tukey’s test was used to detect significant differences (*p* < 0.05) for the in vitro biocontrol assay, and LSD was used to evaluate the in vivo biocontrol assay using grapevine cuttings. To examine normality, the Shapiro–Wilk test was employed, while the Levene test was utilized to assess the homogeneity of variance. In cases where the assumptions were not met, data were appropriately transformed.

## 3. Results

### 3.1. In Vitro Biocontrol Assay (Agar Plug Diffusion Method)

The biocontrol effect by PGPB and endophyte biocontrol bacteria (*Pseudomonas* sp. GcR15a, *Pseudomonas* sp. AMCR2b and *Rhodococcus* sp. PU4) against *D. seriata* isolates (PUCV 2120, PUCV 2142 and PUCV 2183) was evaluated through the agar plug diffusion method. The activity was evaluated using the agar plug diffusion method at different temperatures (Figure 1, Figure 2, Figure 3 and Figure 4), where the fungal mycelial growth inhibition was determined by measuring the inner radius of the fungal growth area.

For the biocontrol by the diffusion method of *D. seriata* PUCV 2120 (Figure 1), strain GcR15a exhibited a significant decrease in growth after 7 and 14 days at 8 and 22 °C (Figure 4a,d), while at 35 °C, no biocontrol effect was observed. Strain GcR15a reached 75% and 56% inhibition after 14 days at 8 and 22 °C, respectively. Strain AMCR2b presented a significant inhibitory effect on growth after 7 and 14 days at 8 and 22 °C (Figure 4a,d). At 35 °C, inhibition was observed only after 14 days (Figure 1c). Strain AMCR2b reached 20%, 65% and 43% inhibition after 14 days at 8, 22 and 35 °C, respectively. With strain AMCR2b, changes in the color of the fungal growth were also observed after 7 and 14 days at 22 °C (Figure 1b). For strain PU4, a significant decrease in *D. seriata* growth was observed after 7 and 14 days only at 8 °C (Figure 4a), reaching 74% inhibition after 14 days. Strain AMCR2b presented the highest decrease in the inner radius of isolate PUCV 2120 (Figure 1). The positive control (tebuconazole) presented complete inhibition after 7 and 14 days at 8 °C (Figure 4a), while at 7 days at 22 °C, there was a significant effect; however, the effect was lost after 14 days (Figure 4d). Significant biocontrol effects were observed at high temperatures only after 14 days at 35 °C (Figure 4g).

For the biocontrol by the diffusion method of *D. seriata* PUCV 2142 (Figure 2), strain GcR15a exhibited a significant decrease in growth after 7 and 14 days at 8 and 22 °C (Figure 4a,d), while at 35 °C, a biocontrol effect was not observed. Strain GcR15a reached 55% and 46% inhibition after 14 days at 8 and 22 °C, respectively. Strain AMCR2b showed a significant inhibitory effect on growth after 14 days at 8 °C (Figure 4b), while at 22 and 35 °C, a decrease was observed after 7 and 14 days (Figure 4e,h). Strain AMCR2b reached 20%, 65% and 43% inhibition after 14 days at 8, 22 and 35 °C, respectively. For strain PU4, a significant decrease at 8 °C was observed after 7 and 14 days (Figure 4b), while at 35 °C, a significant decrease was observed only after 14 days (Figure 1c). In contrast, no effect was observed at 22 °C. Strain PU4 reached 65% and 27% inhibition after 14 days at 8 and 35 °C, respectively. Strain AMCR2b presented the highest decrease in the inner radius of strain PUCV 2142 (Figure 2). The positive control presented a significant inhibitory effect at 8 and 35 °C after 7 and 14 days (Figure 4b,h), while at 22 °C, its effect was significant only after 7 days (Figure 4d).

The biocontrol through the diffusion method of *D. seriata* PUCV 2183 (Figure 3) showed the same effects of strains GcR15a and AMCR2b, significantly decreasing growth after 14 days at 8 °C (Figure 4c) and after 7 and 14 days at 22 and 35 °C (Figure 4f,i). Strain GcR15a reached 40%, 59% and 20% inhibition after 14 days at 8, 22 and 35 °C, respectively. Strain AMCR2b reached 27%, 61% and 31% inhibition after 14 days at 8, 22 and 35 °C, respectively. For strain PU4, a significant decrease at 8 °C and 35 °C was observed after 14 days (Figure 4c,i), while no inhibition was observed at 22 °C. Strain PU4 reached 55% and 35% inhibition after 14 days at 8 and 35 °C, respectively. Strains GcR15a and AMCR2b presented the highest decrease in the inner radius of strain PUCV2183 (Figure 3). The positive control presented complete inhibition after 7 and 14 days at 8 °C (Figure 4c), while at 22 °C, its effect was significant after 7 days (Figure 4f) and at 35 °C after 7 and 14 days (Figure 4i).

From the negative controls, we observed that all *D. seriata* isolates showed higher growth at 22 °C, followed by 35 and 8 °C (Figure 1, Figure 2 and Figure 3), highlighting that the isolate PUCV 2120 presented warped growth at 35 °C with all treatments (Figure 1c). On the other hand, at 22 °C, a noticeable change in the color of the growth (demelanizing effect) of all *D. seriata* strains (PUCV 2120, PUCV 2142 and PUCV 2183) was also observed with the biocontrol strains GcR15a and AMCR2b (Figure 1b, Figure 2b and Figure 3b).

### 3.2. In Vitro Biocontrol Assay (Double Plate Method)

The biocontrol effect from PGPB and endophyte bacteria (*Pseudomonas* sp. GcR15a, *Pseudomonas* sp. AMCR2b and *Rhodococcus* sp. PU4) against *D. seriata* isolates (PUCV 2120, PUCV 2142 and PUCV 2183) was examined through the double plate method at different temperatures (Figure 5, Figure 6, Figure 7 and Figure 8). The fungal mycelial growth inhibition was determined by measuring the diameter of the fungal growth. With respect to the biocontrol of all the *D. seriata* isolates at 22 °C, there was no significant biocontrol effect observed for any strain.

For the biocontrol effect via the double plate method of *D. seriata* PUCV 2120 (Figure 5), strains GcR15a and AMCR2b showed the same effect, significantly decreasing the growth after 7 and 14 days at 8 °C (Figure 8a), while no effect was observed at 35 °C (Figure 8d). For strain PU4, a significant decrease at 8 °C was observed after 14 days (Figure 8a). At 35 °C, a significant decrease was determined after 7 and 14 days (Figure 8d). Strain PU4 presented the highest decrease in the growth diameter of the isolate PUCV 2120 (Figure 5).

Biocontrol assays of *D. seriata* PUCV 2142 with the double plate method (Figure 6) showed no significant decrease in the diameter of the fungal growth with any of the treatments (Figure 8b,e).

For biocontrol through the *D. seriata* PUCV 2183 double plate method (Figure 7), the strains GcR15a and AMCR2b did not present significant decreases in the growth of the fungus at 8 and 35 °C (Figure 8c,f). For strain PU4, only a significant decrease at 35 °C was observed at 14 days of evaluation (Figure 8f), while at 8 °C, no effect was observed (Figure 8d).

In this assay, the bacterial strains studied did not have a significant effect through the double plate method, contrasting with the results observed with the agar plug diffusion method. Similar to the agar plug diffusion assay at 22 °C, a notable color change in the growth and demelanizing effect of isolates PUCV 2142 and PUCV 2183 was observed with strains GcR15a and AMCR2b (Figure 6 and Figure 7). In addition, irregular growth of isolate PUCV 2120 at 35 °C was observed (Figure 7c).

### 3.3. Biocontrol on Grapevine Cuttings

The effects of *D. seriata* isolates on grapevine cuttings pre-inoculated with PGPB and endophyte biocontrol bacteria (*Pseudomonas* sp. GcR15a, *Pseudomonas* sp. AMCR2b and *Rhodococcus* sp. PU4, respectively) at 8 and 22 °C were evaluated. The damage to the grapevine cuttings was observed by measuring the vascular lesion length after two months (Appendix A).

For the biocontrol effect of *D. seriata* PUCV 2120 (Figure 9), only strain AMCR2b showed a significant decrease in the vascular lesion length at 8 and 22 °C with respect to the negative control (Figure 9a,d).

For the biocontrol effect of *D. seriata* PUCV 2142 (Figure 9), none of the treatments exhibited significant differences with respect to the negative control at 8 and 22 °C (Figure 9b,e).

For the biocontrol effect of *D. seriata* PUCV 2183 (Figure 9), at 22 °C, strains AMCR2b and PU4 showed a significant decrease in the vascular lesion length with respect to the negative control (Figure 9e), while none of the treatments presented significant differences with respect to the negative control at 8 °C (Figure 9c).

The control percentages on *D. seriata* after 14 days of each bacterium in the different tests were obtained using Formula (1), indicated above and shown in Table 1. The best results were obtained by the in vitro assay using the agar plug diffusion method at a temperature of 8 °C, where *Rhodococcus* sp. PU4 obtains a 65% control (average) and *Pseudomonas* sp. GcR15a, a 57% average. On the other hand, at 22 °C, only strains of *Pseudomonas* sp. showed control (greater than 50%). At 35 °C, *Pseudomonas* AMCR2b showed higher control (38%), on average, similar to the positive control tebuconazole (33%) and *Rhodococcus* sp. (30%).

## 4. Discussion

*Diplodia seriata* is one of the most frequently isolated fungal species from diseased grapevines in different countries, such as Chile, USA (California), Mexico, Spain, Portugal, France, Iran, Lebanon, Algeria, Tunisia and Australia [4,13,14,15,16,17,18,19,20,21,22,23,24,25,26]. Using microorganisms for the biocontrol of *D. seriata* is a promising alternative to chemical fungicides [30,31,32]. Due to the wide temperature growth range of *D. seriata* [22,27], its biocontrol should be evaluated at different temperatures. This study assessed the capability of two bacteria of the genus *Pseudomonas* (GcR15a and AMCR2b) and one of *Rhodococcus* (PU4) to inhibit the mycelial growth of three Chilean *D. seriata* isolates (PUCV 2120, PUCV 2142 and PUCV 2183) at 8, 22 and 35 °C. Some works evaluated the in vitro growth of biocontrol bacteria at different temperatures, but the biocontrol with pathogenic fungi (including *D. seriata*) effect was only at 25 °C [45]. Also, antagonistic effects against phytopathogenic fungi at different temperatures have rarely been reported [46,47,48,49].

The best results were obtained by using the agar plug diffusion method, where up to 75% inhibition of mycelial growth was obtained with *Pseudomonas* sp. (GcR15a) at a temperature of 8 °C. In comparison, the double plate method’s maximum efficiency was 42% with *Rhodococcus* sp. at 35 °C (Table 1). Alvarez-Pérez et al. [33] observed inhibition by *Streptomyces* and *Saccharopopolys* strains against *D. seriata* CBS 112555, between 29% and 61% after 12 days of incubation at 25 °C. On the other hand, Niem et al. [31] observed the inhibition of mycelial growth (12–47%) with *Pseudomonas* strains against *D. seriata* (isolate A142a) after 7 days of incubation at 25 °C, results similar to those obtained in this study (Figure 4d–f). Silva-Valderrama et al. [32] observed inhibition in the growth of *D. seriata* (isolate 117 Molina) 15–100% after 21 days of incubation at 25 °C, in this case using biocontrol fungi, such as *Chaetomium* sp., *Cladosporium* sp., *Clonostachys*
*rosea*, *Epicoccum*
*nigrum*, *Purpureocillium lilacinum* and *Trichoderma* sp. An important aspect is that the biocontrol work with *D. seriata* described above uses one isolate. Our results highlight the importance of considering more than one when evaluating the effectiveness of a biocontroller. Previous reports support the importance of evaluating the effect of different *D. seriata* isolates that presented different mycelial growth at 22 °C and differences in necrosis length in detached canes and potted vines [50].

Regarding reports at different temperatures, in a temperature range of 25 to 35 °C, *Pseudomonas fluorescens* RG-26 inhibits the growth (≥50%) of *Fusarium oxysporum* f. sp. *ciceris* race 5, whereas the *Trichoderma* sp. isolate Td-1 inhibited (36–56%) the growth of *Sclerotium rolfsii* [46]. Guetsky et al. [47] and Manaa et al. [49] highlighted the biocontrol effect at temperatures ≤10 °C, observing a significant decrease in fungus *Aspergillus flavus* KCCM 60330 growth with the bacterium *Pseudomonas protegens* AS15 at 10, 20, 30 and 40 °C [49], and a similar effect was reported against *B. cinerea* with a bacterial consortium composed of *Pichia guilermondii* and *Bacillus mycoides* at 4, 10, 15, 20, 25, 30 and 36 °C, with spore germination inhibition between 20% and 80% [47]. Knowing biocontrol at low temperatures is important for the protection of exposed wounds of vine plants generated by pruning during winter [51], which is consistent with most dispersion times of spores of causal fungi of dieback-type trunk diseases in Mediterranean climate zones such as Chile [15] and California [16,52]. In addition, the importance of demonstrating the biocontrol effect at different temperatures is in accordance with increasing evidence that suggests that extreme temperatures (cold or heat) will increase in the coming years [28]. Therefore, microbial products that alleviate the stress caused by climate change in crops are required for agriculture [53,54].

In this study, after seven days, tebuconazole inhibited mycelial growth of isolates of *D. seriata* in PDA agar plates 100% at 8 °C, 42–48% at 22 °C and 35–50% at 35 °C, maintaining an inhibition at 14 days of 88–100% to 8 °C, 0–12% at 22 °C and 15–48% at 35 °C. Therefore, tebuconazole presented high efficacy at 8 °C. A tebuconazole-based treatment was able to inhibit the conidial and mycelial germination of three *D. seriata* isolates (Vid 1472, Vid 1468 and Vid 1270) at temperatures of 22 and 25 °C [37]. Treatments with *Pseudomonas* strains GcR15a and AMCR2b at 22 °C maintained biocontrol for 14 days for the three *D. seriata* isolates. This study highlighted the importance of carrying out tests on plants naturally infected or inoculated with *D. seriata* conidia to determine how infection occurs in nature [55,56].

In the present work, we observed that the greatest biocontrol effect was exhibited by *Pseudomonas* strains, which were isolated from cold environments [35]. The biocontrol potential of *Pseudomonas* strains against phytopathogens has been previously reported [31,35,50,57,58,59,60,61]. The double plate assay method suggested that the biocontrol effect exhibited by the bacteria is probably produced by diffusible compounds [45]. In accordance, diffusible compounds were more effective than volatile compounds in in vitro assays using *Pseudomonas* strains against *B. cinerea* [57]. Niem et al. [31] showed that *Pseudomonas* is a healthy vineyard’s predominant endophytic bacterial genus. In contrast, this genus decreased significantly in diseased plants with GTDs, suggesting a possible effect of *Pseudomonas* strains against *D. seriata* and other fungi associated with GTDs. On the other hand, the biocontrol potential of *Rhodococcus* spp. against phytopathogens has rarely been reported [62,63]. In our work, *Rhodococcus* sp. PU4 presented a significant inhibitory effect through the agar plug method in some cases. Still, its inhibitory effect on the growth of *D. seriata* strains PUCV 2120 and PUCV 2183 through the double plate method at high temperatures (35 °C) was higher than the effect of *Pseudomonas* strains. In a study by De Oliveira et al. [64], they evaluated the effectiveness of 70 actinomycete isolates with other pathogens at three different temperatures: 25, 30 and 35 °C. The best antagonistic bacteria had their effectiveness maximized at 30 °C. In our work, the actinobacteria *Rhodococcus* performed at 8 °C and 35 °C, while at 22 °C, this microorganism could not contain the *D. seriata* growth.

A potential bacterial consortium composed of these three bacteria seems attractive for future studies, presenting various biocontrol mechanisms against *D. seriata*. The strategy of consortia may increase the efficacy and improve the biocontrol effects due to synergistic mechanisms [60].

In some cases, in in vitro assays, no growth inhibition but just a change in color was observed for *D. seriata* growth by the biocontrol bacteria. A darker phenotype was observed in the control at longer incubation times at the optimal temperature (22 °C) and 35 °C, possibly due to melanin production. Genes involved in melanin syntheses, such as DOPA-melanin (production of aerial mycelium and protection against enzymatic lysis and oxidative stress), DHN-melanin (ramification of mycelium when exposed to nutrient deficiency) and pyomelanin (hyphae development), were conserved among *Botryosphariaceae*, highlighting the importance of melanin in pathogenesis [65]. Melanization is not an essential factor for fungal growth but contributes to the survival of cells under environmental stress conditions and may confer virulence in pathogens [65,66]. Demelanizing activity was observed with *Ganoderma lucidum* extracts against *Aspergillus niger*, indicating that the decrease in pigmentation could reduce this virulence factor [67].

When considering the results obtained in vine cuttings, there were differences in biocontrol at 8 and 22 °C depending on the type of *D. seriata* isolate used. *Pseudomonas* sp. AMCR2b presented a biocontrol effect against the isolate PUCV 2120 at both temperatures and against the isolate PUCV 2183 only at 22 °C, and *Rhodococcus* sp. PU4 showed a biocontrol effect only against the isolate PUCV 2183 at 22 °C, while none of the bacterial strains presented activity against the isolate PUCV 2142, demonstrating that it is a *D. seriata* isolate more resistant to control, even by the fungicide tebuconazole. Previous studies performed by Larach et al. [6] showed that the isolate PUCV 2142 presented a greater lesion diameter on grape berries than isolates PUCV 2120 and PUCV 2183. This aspect reinforces the need to evaluate more than one isolate per pathogen, an aspect not previously considered by some authors [31,32,33]. Therefore, the biocontrol effects of the bacteria studied in this work could prevent the fungus from establishing itself in grapevine plants through direct inhibition or demelanization. In future trials, it will be interesting to test the biocontrol effects of these bacteria in plants with a mixture of *D. seriata* isolates at different temperatures.

## 5. Conclusions

In this study, the growth inhibition of three *D. seriata* isolates by beneficial bacteria was observed. The highest inhibition was observed with both *Pseudomonas* strains using the agar plug diffusion assay at different temperatures. In assays without fungal inhibition, antifungal demelanizing activity was observed. *Rhodococcus* sp. PU4 showed biocontrol potential by producing potential VOCs capable of inhibiting *D. seriata* growth at high temperatures (35 °C). The in vivo biocontrol assay with cuttings corroborated the biocontrol effect of *Pseudomonas* sp. AMRC2b in two isolates from *D. seriata*, while *Rhodococcus* sp. PU4 presented a biocontrol effect in only one isolate. None of the bacterial strains evaluated in the in vivo biocontrol assay showed an effect against the isolated PUCV 2142, proving to be a highly resistant isolate to control. The study of different pathogen isolates should be mostly evaluated to better understand their different behaviors and to improve their control and biocontrol strategies. The results obtained at low temperatures in in vitro and in vivo assays are promising for effective control in the field, where infections occur mainly in winter.

## Figures and Tables

**Figure 1 microorganisms-12-00350-f001:**
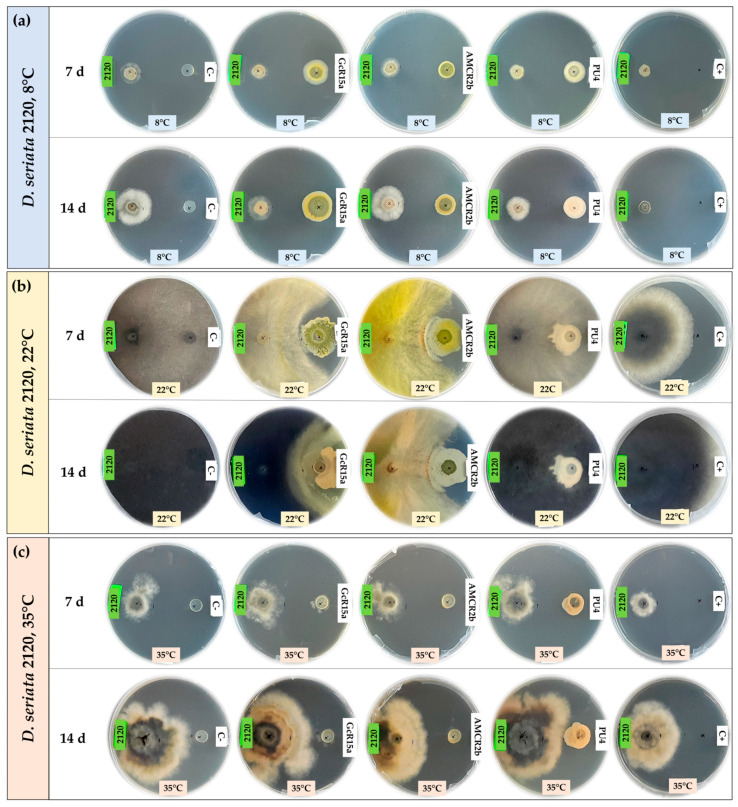
Biocontrol by PGPB and endophyte bacteria against *Diplodia seriata* PUCV 2120 by the agar plug diffusion method at different temperatures. (**a**) Biocontrol against *D. seriata* PUCV 2120 at 8 °C after 7 and 14 days. (**b**) Biocontrol against *D. seriata* PUCV 2120 at 22 °C after 7 and 14 days. (**c**) Biocontrol *D. seriata* PUCV 2120 at 35 °C after 7 and 14 days. Abbreviations: 2120, *D. seriata* PUCV 2120; C−, negative control; GcR15a, *Pseudomonas* sp. GcR15a; AMCR2b, *Pseudomonas* sp. AMCR2b; PU4, *Rhodococcus* sp. PU4; C+, positive control (tebuconazole).

**Figure 2 microorganisms-12-00350-f002:**
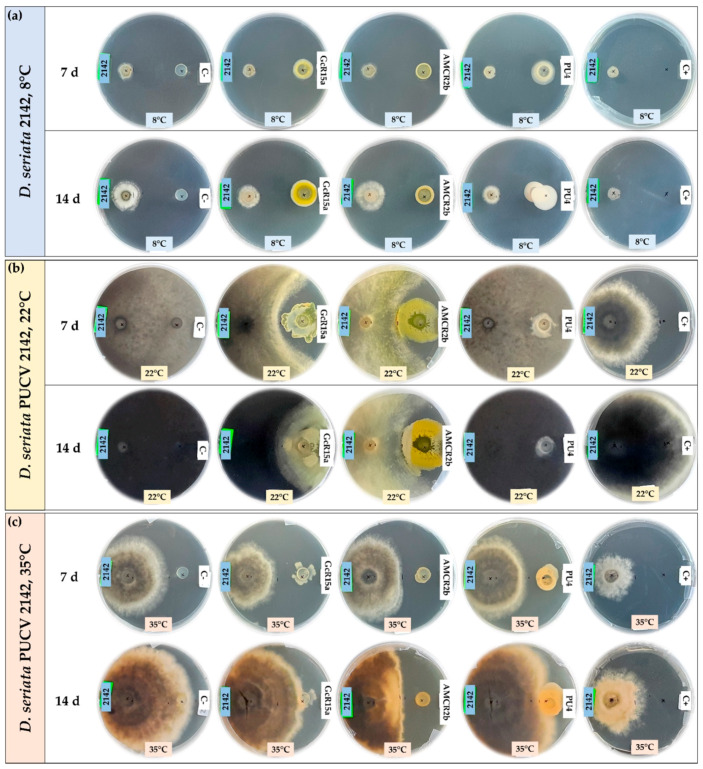
Biocontrol by PGPB and endophyte bacteria against *Diplodia seriata* PUCV 2142 by the agar plug diffusion method at different temperatures. (**a**) Biocontrol against *D. seriata* PUCV 2142 at 8 °C after 7 and 14 days. (**b**) Biocontrol against *D. seriata* PUCV 2142 at 22 °C after 7 and 14 days. (**c**) Biocontrol against *D. seriata* PUCV 2142 at 35 °C after 7 and 14 days. Abbreviations: 2142, *D. seriata* PUCV 2142; C−, negative control; GcR15a, *Pseudomonas* sp. GcR15a; AMCR2b, *Pseudomonas* sp. AMCR2b; PU4, *Rhodococcus* sp. PU4; C+, positive control (tebuconazole).

**Figure 3 microorganisms-12-00350-f003:**
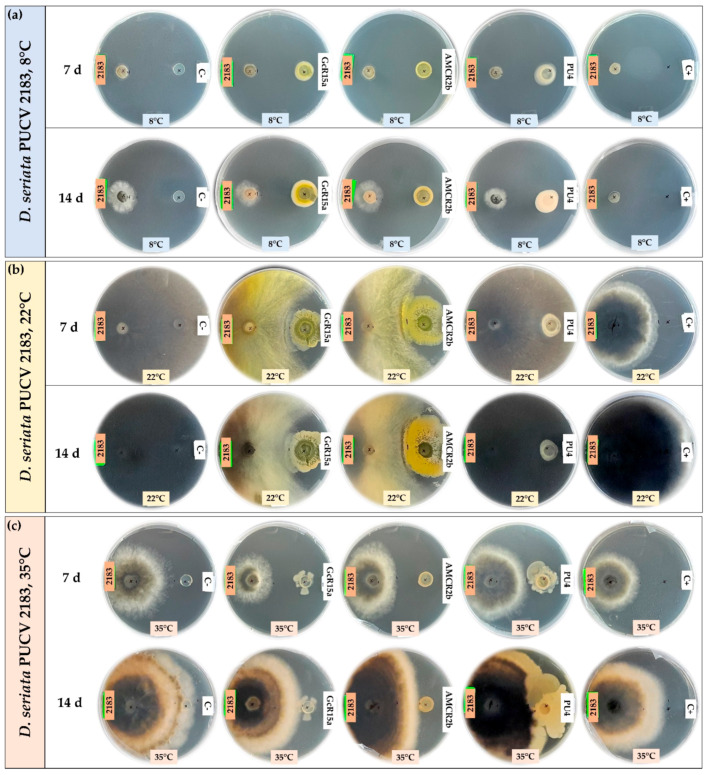
Biocontrol by PGPB and endophyte bacteria against *Diplodia seriata* PUCV 2183 by the agar plug diffusion method at different temperatures. (**a**) Biocontrol against *D. seriata* PUCV 2183 at 8 °C after 7 and 14 days. (**b**) Biocontrol against *D. seriata* PUCV 2183 at 22 °C after 7 and 14 days. (**c**) Biocontrol against *D. seriata* PUCV 2183 at 35 °C after 7 and 14 days. Abbreviations: 2183, *D. seriata* PUCV 2183; C−, negative control; GcR15a, *Pseudomonas* sp. GcR15a; AMCR2b, *Pseudomonas* sp. AMCR2b; PU4, *Rhodococcus* sp. PU4; C+, positive control (tebuconazole).

**Figure 4 microorganisms-12-00350-f004:**
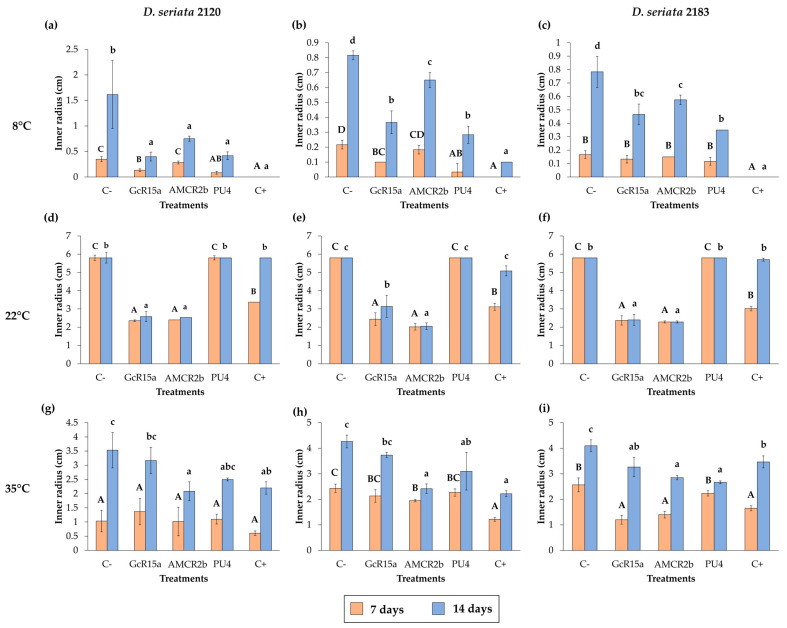
Effects of PGPB and endophyte bacteria on the inner radius of *Diplodia seriata* isolates by the agar plug diffusion method at different temperatures. (**a**–**c**) Effects on the inner radius of the isolates of *D. seriata* at 8 °C after 7 and 14 days. (**d**–**f**) Effects on the inner radius of *D. seriata* isolates at 22 °C after 7 and 14 days. (**g**–**i**) Effects on the inner radius of *D. seriata* isolates at 35 °C after 7 and 14 days. Means with different letters indicate significant differences (*p* < 0.05) and uppercase and lowercase letters correspond to 7 and 14 days, respectively. Abbreviations: C−, negative control; GcR15a, *Pseudomonas* sp. GcR15a; AMCR2b, *Pseudomonas* sp. AMCR2b; PU4, *Rhodococcus* sp. PU4; C+, positive control (tebuconazole).

**Figure 5 microorganisms-12-00350-f005:**
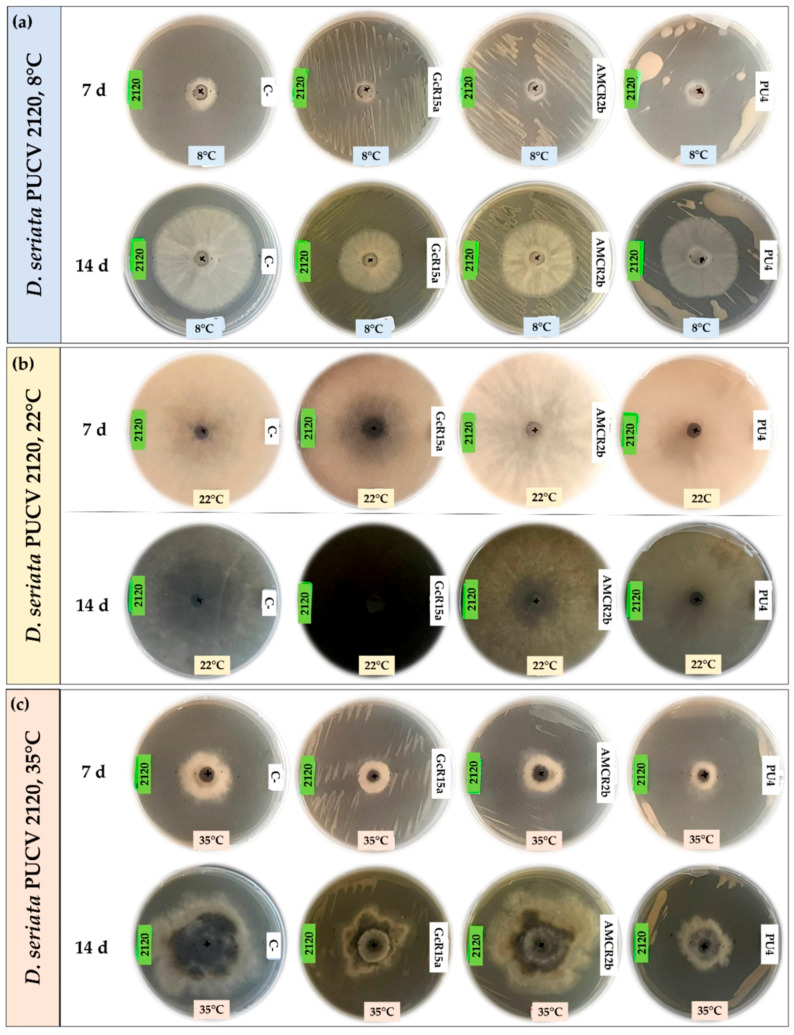
Biocontrol by PGPB and endophyte bacteria against *Diplodia seriata* PUCV 2120 by the double plate method at different temperatures. (**a**) Biocontrol against *D. seriata* PUCV 2120 at 8 °C after 7 and 14 days. (**b**) Biocontrol against *D. seriata* PUCV 2120 at 22 °C after 7 and 14 days. (**c**) Biocontrol against *D. seriata* PUCV 2120 at 35 °C after 7 and 14 days. Abbreviations: 2120, *D. seriata* PUCV 2120; C−, negative control; GcR15a, *Pseudomonas* sp. GcR15a; AMCR2b, *Pseudomonas* sp. AMCR2b; PU4, *Rhodococcus* sp. PU4.

**Figure 6 microorganisms-12-00350-f006:**
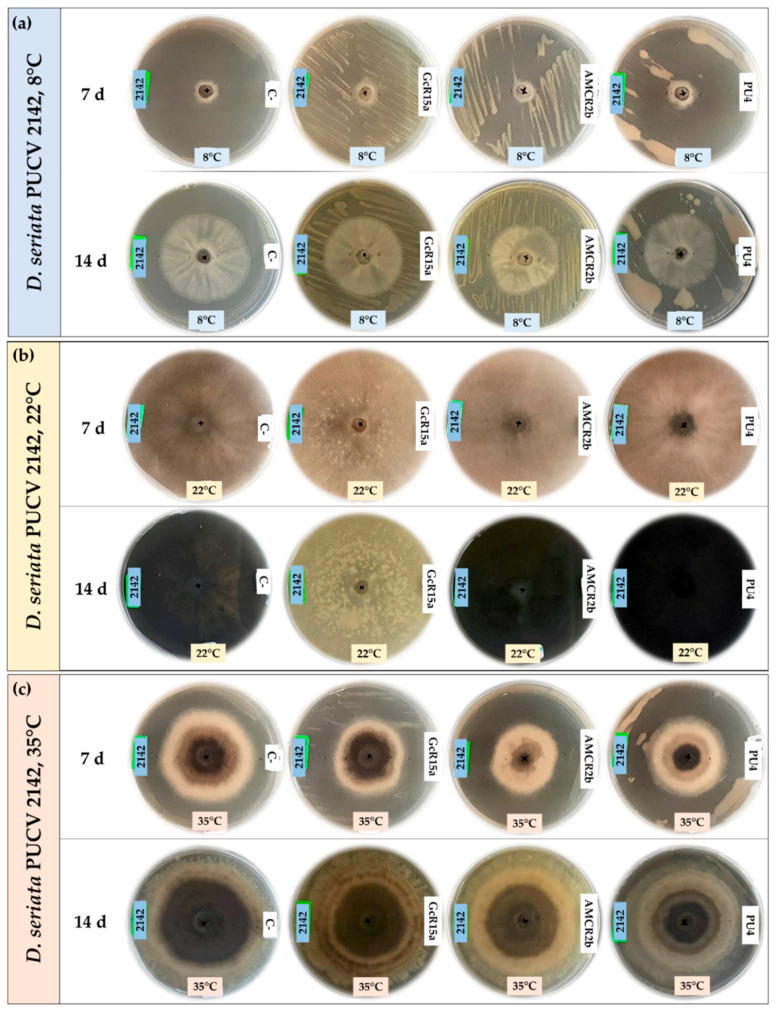
Biocontrol by PGPB and endophyte bacteria against *Diplodia seriata* PUCV 2142 by the double plate method at different temperatures. (**a**) Biocontrol against *D. seriata* PUCV 2142 at 8 °C after 7 and 14 days. (**b**) Biocontrol against *D. seriata* PUCV 2142 at 22 °C after 7 and 14 days. (**c**) Biocontrol against *D. seriata* PUCV 2142 at 35 °C after 7 and 14 days. Abbreviations: 2142, *D. seriata* PUCV 2142; C−, negative control; GcR15a, *Pseudomonas* sp. GcR15a; AMCR2b, *Pseudomonas* sp. AMCR2b; PU4, *Rhodococcus* sp. PU4.

**Figure 7 microorganisms-12-00350-f007:**
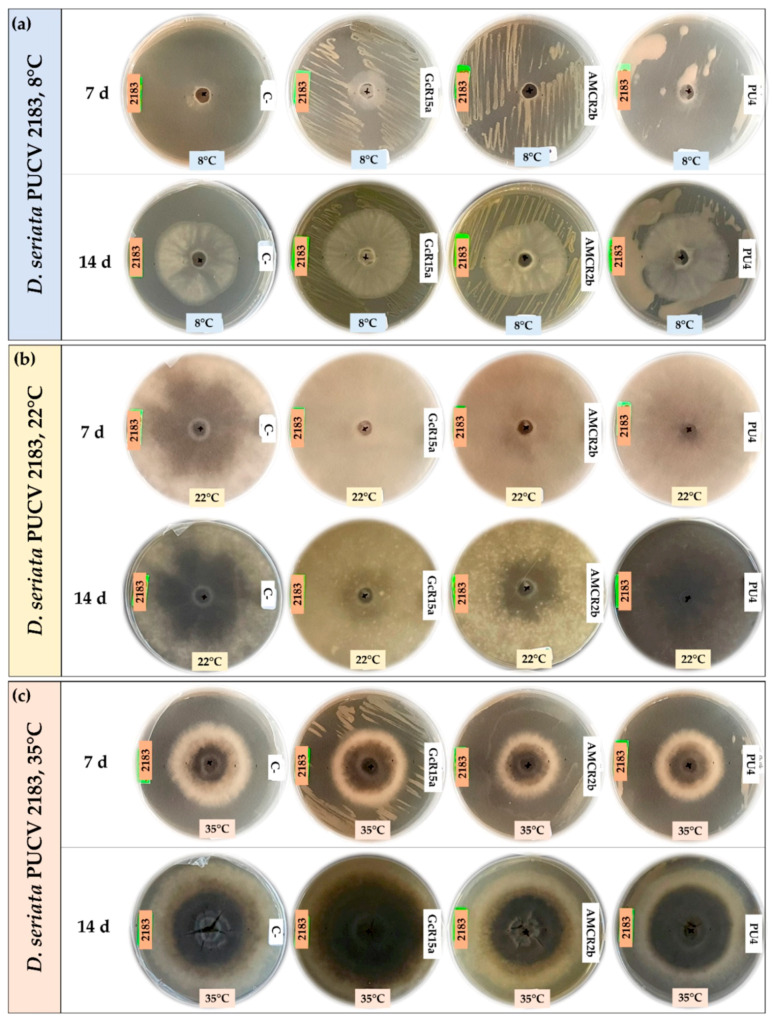
Biocontrol by PGPB and endophyte bacteria against *Diplodia seriata* PUCV 2183 by the double plate method at different temperatures. (**a**) Biocontrol against *D. seriata* PUCV 2183 at 8 °C after 7 and 14 days. (**b**) Biocontrol against *D. seriata* PUCV 2183 at 22 °C after 7 and 14 days. (**c**) Biocontrol against *D. seriata* PUCV 2183 at 35 °C after 7 and 14 days. Abbreviations: 2183, *D. seriata* PUCV 2183; C−, negative control; GcR15a, *Pseudomonas* sp. GcR15a; AMCR2b, *Pseudomonas* sp. AMCR2b; PU4, *Rhodococcus* sp. PU4.

**Figure 8 microorganisms-12-00350-f008:**
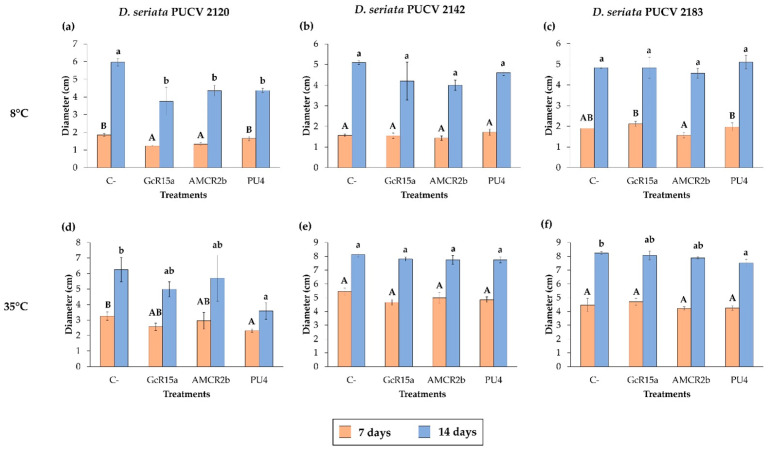
Effects of PGPB and endophyte bacteria on the diameter of *Diplodia seriata* isolates by the double plate method at different temperatures. (**a**–**c**) Effects on the diameter of the isolates of *D. seriata* at 8 °C after 7 and 14 days. (**d**–**f**) Effects on the inner radius of the isolates of *D. seriata* at 35 °C after 7 and 14 days. There was no decrease in the diameter of the isolates of *D. seriata* with any treatments at 22 °C and the fungi grew entirely from 7 days (8.5 cm). Means with different letters indicate significant differences (*p* < 0.05) and uppercase and lowercase letters correspond to 7 and 14 days, respectively. Abbreviations: C−, negative control; GcR15a, *Pseudomonas* sp. GcR15a; AMCR2b, *Pseudomonas* sp. AMCR2b; PU4, *Rhodococcus* sp. PU4.

**Figure 9 microorganisms-12-00350-f009:**
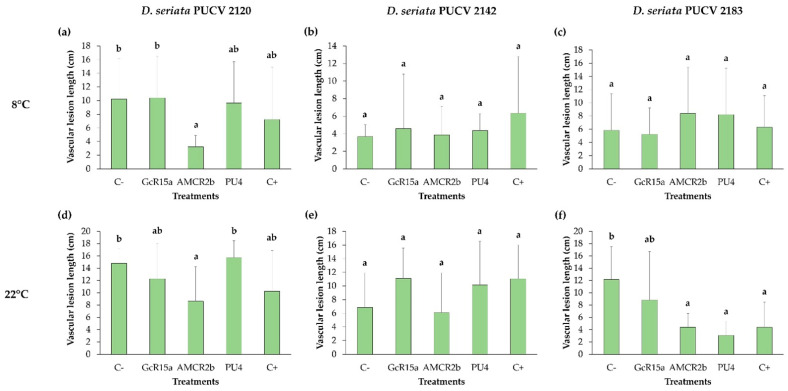
Effects of *Diplodia seriata* isolates grapevine pruning material pre-inoculated with PGPB and endophyte bacteria at different temperatures. (**a**–**c**) Effects of pre-inoculation with native bacteria on the vascular lesion length of grapevine pruning material inoculated with *D. seriata* isolates at 8 °C. (**d**–**f**) Effects of pre-inoculation with native bacteria on the vascular lesion length of grapevine pruning material inoculated with *D. seriata* isolates at 22 °C. Means with different letters indicate significant differences (*p* < 0.05). Abbreviations: C−, negative control; GcR15a, *Pseudomonas* sp. GcR15a; AMCR2b, *Pseudomonas* sp. AMCR2b; PU4, *Rhodococcus* sp. PU4; C+, positive control (tebuconazole).

**Table 1 microorganisms-12-00350-t001:** Percentage of inhibition of *D. seriata* isolates by PGPB and endophyte bacteria at different temperatures at 14 days of confrontation.

	T °C	Pathogen	Inhibition Percentage (%)			
			C+	GcR15a	AMCR2b	PU4
Agar plug diffusion (in vitro)	8 °C	*D. seriata* 2120	100	75	53	74
		*D. seriata* 2142	88	55	20	65
		*D. seriata* 2183	100	40	26	55
	22 °C	*D. seriata* 2120	0	56	56	0
		*D. seriata* 2142	12	45	64	0
		*D. seriata* 2183	2	58	60	0
	35 °C	*D. seriata* 2120	37	10	41	29
		*D. seriata* 2142	48	12	43	27
		*D. seriata* 2183	15	20	30	34
Double plate (in vitro)	8 °C	*D. seriata* 2120	-	37	27	27
		*D. seriata* 2142	-	17	21	10
		*D. seriata* 2183	-	0	5	0
	22 °C	*D. seriata* 2120	-	0	0	0
		*D. seriata* 2142	-	0	0	0
		*D. seriata* 2183	-	0	0	0
	35 °C	*D. seriata* 2120	-	20	9	42
		*D. seriata* 2142	-	4	5	5
		*D. seriata* 2183	-	2	4	9
Cuttings (in vivo)	8 °C	*D. seriata* 2120	29	0	68	5
		*D. seriata* 2142	0	0	0	0
		*D. seriata* 2183	0	10	0	0
	22 °C	*D. seriata* 2120	31	17	42	0
		*D. seriata* 2142	0	0	10	0
		*D. seriata* 2183	64	27	63	75

Abbreviations: C+, positive control (tebuconazole). GcR15a, *Pseudomonas* sp. GcR15a; AMCR2b, *Pseudomonas* sp.; PU4, *Rhodococcus* sp. PU4.

## Data Availability

The partial sequences of the ITS and β-tub genes of *D. seriata* PUCV 2120, *D. seriata* PUCV 2142 and *D. seriata* PUCV 2183 are deposited in GenBank accession number MT023573-MT063140; MT023574-MT063141 and MT023587-MT063154, respectively for each isolate. The partial 16S rRNA gene sequences of *Pseudomonas* sp. GcR15a, *Pseudomonas* sp. AMCR2b and *Rhodococcus* sp. PU4 are deposited in GenBank under the accession numbers MW548343, OQ244037 and OQ244039, respectively.

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
