# Peer review of "Diplodia seriata* Biocontrol Is Altered via Temperature and the Control of Bacteria"

_microorganisms, 2024, doi:10.3390/microorganisms12020350_

Round 1
Reviewer 1 Report
Comments and Suggestions for Authors
Dear Editor/Author,
The results of the manuscripts are valuable for science, however, I do have some concerns that I believe would benefit from further attention before publication.
1. There is lack of the accurate identification in this study. Why the bacteria has not been identified at the species level?
2. The authors keep using “native bacteria” throughout the manuscript that is absolutely misleading. The author should know that pollution genetic studies need to be conducted for different strains to find out if a microorganism is native or invasive!
3. Section “2.3 In vitro biocontrol assay of diffusible organic compounds” needs to be replaced with “Dual culture antagonism assay”.
4. Section “2.4 In vitro biocontrol assay of volatile organic compounds” is not correct. First the methods of streaking is not correct and it should have use the different serial dilution. Secondly, the method is simply different method for dual culture antagonism assay and there is no volatile study is involved. I would suggest the author read the other studies that have done volatile compounds assays.
5. Section 2.5: It is unclear that how many cut branches were used. Also the branches had been tested for absent/present of other endophytic fungi/bacteria prior the assay?
6. Pictures of 2.5 needs to be added.
7. Discussion is not strong enough, most of results have been repeated there while need to discuss the outcome and compare it with the other studies.
Please see more comments on the text.

Author Response
Reviewer 1
- 1. There is lack of accurate identification in this study. Why the bacteria has not been identified at the species level?
Q: The bacteria used in this work are in the process of obtaining a patent of use. Patent request: PCT/CL2022/050057, so at this moment, we prefer to use Pseudomonas sp.
- The authors keep using “native bacteria” throughout the manuscript that is absolutely misleading. The author should know that pollution genetic studies need to be conducted for different strains to find out if a microorganism is native or invasive!
Q: We consider what was indicated, so instead of native, we will call it PGPB, according to the work done by Vega-Celedón et al. 2021. Microorganisms 2021, 9, 538. https://doi.org/10.3390/microorganisms9030538. In addition, we indicate which are PGPB and which are vine endophytic bacteria.
- Section “2.3 In vitro biocontrol assay of diffusible organic compounds” needs to be replaced with “Dual culture antagonism assay”.
Q: The change was done. However, we need to differentiate between both dual culture antagonism assays. In this assay, we changed the title to "Effect of diffusible compounds produced by bacteria." The reference was Olivera et al. 2021. Microorganisms 2021, 10, 664. https://doi.org/10.3390/antibiotics10060664
- Section “2.4 In vitro biocontrol assay of volatile organic compounds” is not correct. First the methods of streaking is not correct and it should have use the different serial dilution. Secondly, the method is simply different method for dual culture antagonism assay and there is no volatile study is involved. I would suggest the author read the other studies that have done volatile compounds assays.
Q: The change was done following the same criteria as 2.3. The title was left as "In vitro effect of volatile compounds produced by bacteria". The reference was Delgado et al. 2021. Antibiotics 10(6): 663. doi: 10.3390/antibiotics10060663. PMID: 34205962. Both microorganisms never get in touch, so only volatile compounds can affect the growth of Diplodia seriata. this methodology was also described by Gao et al. 2017 https://doi.org/10.1016/j.biocontrol.2016.11.007 In this last article authors describe the use of a bacteria employing the same methodology used in this paper.
- Section 2.5: It is unclear that how many cut branches were used. Also, the branches had been tested for absent/present of other endophytic fungi/bacteria prior the assay?
Q: Six cuttings were used for each treatment, isolate and temperature, a total of 270 cuttings; we added these details in the text. In this trial, we include five treatments, three different temperatures, three isolates, and six cuttings, a total of 270.
We did not test the cuttings for endophytic fungi/bacteria, we only leave the respective inoculated controls and positive control. With these treatments we could differentiate between no biocontrol agent and the respective bacteria isolate in comparison with a commercial treatment.
- Pictures of 2.5 needs to be added.
Q: We add a Figure in supplementary material (FigureS1).
- Discussion is not strong enough, most of results have been repeated there while need to discuss the outcome and compare it with the other studies.
In the discussion chapter, we discuss our results with two new papers. However, no further information is available on the biocontrol of Botryosphaeriaceae at different temperatures.
Additionally, we can say that the cuttings were kept at 5°C for two weeks before use: since they were harvested at the end of summer, when they were semi-lignified, and preserved until inoculation, preventing thus from decomposing.
Reviewer 2 Report
Comments and Suggestions for Authors
Comments:
To the Editor Prof.
The manuscript “The In Vitro and In Vivo Biocontrol Capacity of Diplodia seriata Is Altered by Temperature and the Controlling Bacteria’’ describes the biocontrol effects of three native bacteria, Pseudomonas sp. GcR15a, Pseudomonas sp. AMCR2b, and Rhodococcus sp. PU4, on three D. seriata isolates at 8, 22, and 35°C. The study is interesting and of great practical importance. The experimental studies are mostly carried out professionally. The article satisfies the criteria of the Microorganisms. I recommended that this paper be accepted after minor revision.
1. Arrange keywords alphabetically.
2. Materials and Methods, Three phytopathogenic D. seriata isolates were used in this study: PUCV 2120, PUCV 2142, and PUCV 2183, obtained from diseased grapevine plants [4]. Why three isolates?
3. Materials and Methods, In vivo biocontrol assay. Why the cuttings did not grow in pots?
With best regards
Author Response
Reviewer 2
The manuscript “The In Vitro and In Vivo Biocontrol Capacity of Diplodia seriata Is Altered by Temperature and the Controlling Bacteria’’ describes the biocontrol effects of three native bacteria, Pseudomonas sp. GcR15a, Pseudomonas sp. AMCR2b, and Rhodococcus sp. PU4, on three D. seriata isolates at 8, 22, and 35°C. The study is interesting and of great practical importance. The experimental studies are mostly carried out professionally. The article satisfies the criteria of the Microorganisms. I recommended that this paper be accepted after minor revision.
- Arrange keywords alphabetically.
Q: The arrange was done.
- Materials and Methods, three phytopathogenic D. seriata isolates were used in this study: PUCV 2120, PUCV 2142, and PUCV 2183, obtained from diseased grapevine plants [4]. Why three isolates?
Q: Three isolates were used, one of each studied region, and the three were from different localities: Palmilla, O'Higgins (PUCV2120); Pencahue Maule region (PUCV 2183), and the last from Batuco Maule region (PUCV 2142). These isolates also have different pathogenic abilities', see Larach et al., 2023. Plants 2023(12): 2984. https://doi.org/10.3390/plants12162984
- Materials and Methods, In vivo biocontrol assay. Why the cuttings did not grow in pots?
Q: We used cuttings in order to make an in vivo trial with the same temperatures used in vitro. This was only possible using cuttings. Now we are performing trials with rooted plants but with different treatments and species under field conditions. In one more year we can compare these results. But in these trials, we can not control temperature, only see the performance under natural conditions.
Reviewer 3 Report
Comments and Suggestions for Authors
General comments:
This is an interesting study concerning the influence of temperature on the biocontrol activity against Diplodia seriata. The authors tested the activity of potential biocontrol strains both by volatile and diffusible compounds and tested the effect both in vitro and in vivo. The study subject is economically significant. The introduction provides the necessary background, and the methods are well-planned. I would appreciate the authors adding the table with inhibition values to the manuscript as it would be easier to follow. I would also appreciate it if the concussion would also be accompanied by a summarizing table of graphs, pinpointing how the in vitro activity translates to in vivo results. It’s an extremely important issue. The hyphae color changes are an interesting observation that could be measured and evaluated further. I would like to suggest the authors toad row data to prepare graphs for the supplementary materials. Generally, I consider this an interesting manuscript acceptable for publication after minor revisions.
Please find additional comments below:
In text comments:
Line 2: Please move seriata to the next line (please avoid separating words in titles)
Line 16: Cabernet Sauvignon is one of the most economically important red wine varieties
Line 34: Vitis vinifera L.
Line 51: please rephrase e.g.: is at the leading position amongst…
Line 75: Please provide the access date
Line 107: patent application (but wild-type bacteria cannot be protected by patent application only their use if that is so please refer to the patent application number
Line 163: Please indicate how the assumptions for ANOVA were verified.
If I may suggest your future studies concerning biocontrol consortia it would be interesting to test in vitro interactions between tested strains and the activity in vivo.
Comments on the Quality of English LanguageOnly minor issues, results are a little hard to follow, but the summarising table should solve it.
Author Response
Reviewer 3
This is an interesting study concerning the influence of temperature on the biocontrol activity against Diplodia seriata. The authors tested the activity of potential biocontrol strains both by volatile and diffusible compounds and tested the effect both in vitro and in vivo. The study subject is economically significant. The introduction provides the necessary background, and the methods are well-planned. I would appreciate the authors adding the table with inhibition values to the manuscript as it would be easier to follow. I would also appreciate it if the concussion would also be accompanied by a summarizing table of graphs, pinpointing how the in vitro activity translates to in vivo results. It’s an extremely important issue. The hyphae color changes are an interesting observation that could be measured and evaluated further. I would like to suggest the authors toad row data to prepare graphs for the supplementary materials. Generally, I consider this an interesting manuscript acceptable for publication after minor revisions.
Q: We add the table with inhibitions in the text, as Table 1.
In text comments:
Line 2: Please move seriata to the next line (please avoid separating words in titles)
Q: This suggestion was done.
Line 16: Cabernet Sauvignon is one of the most economically important red wine varieties
Q: This suggestion was done.
Line 34: Vitis vinifera L.
Q: The change was done.
Line 51: please rephrase e.g.: is at the leading position amongst…
Q: This suggestion was done.
Line 75: Please provide the access date
Q: This suggestion was done. We add a new reference.
Line 107: patent application (but wild-type bacteria cannot be protected by patent application only their use if that is so, please refer to the patent application number
Q: We add the patent request: PCT/CL2022/050057
Line 163: Please indicate how the assumptions for ANOVA were verified.
Q: To examine normality, the Shapiro-Wilk test was employed, while the Levene test was utilized to assess the homogeneity of variance. In cases where the assumptions were not met, data were appropriately transformed. We add this to the text.
If I may suggest your future studies concerning biocontrol consortia it would be interesting to test in vitro interactions between tested strains and the activity in vivo.
Q: Thanks for your suggestion we will keep it in mind.
Round 2
Reviewer 1 Report
Comments and Suggestions for Authors
Dear Editor/Author,
I do not see much improvements in the current revised manuscript, except that the author has changed some words.
In this study there is no extraction, measurement and evaluation of any volatile compounds, therefore both new title are incorrect. You cannot give those tiles and assume that there was a compound. The experiments of sections 2.3 and 2.4 are simply the same (dual cultures), just used different methods.
There is no improvement in discussion as still lots of part of results have been repeated there.
Author Response
Dear Editor
We are pleased to send you an improved version of the manuscript titled “The in vitro and in vivo biocontrol capacity of Diplodia seriata is altered by temperature and controlling bacteria”. According to your observation, the proposed title for our work remains “Diplodia seriata biocontrol is altered by temperature and controlling bacteria”.
On the next page, we describe the changes introduced in the new version of the manuscript to respond to the second review of reviewer 1.
We would like to thank the reviewers for their efforts and their contribution to improving this work.
We hope that the changes made are appropriate for the publication of this article in this prestigious journal.
Sincerely,
Ximena Besoain, Dra.
Full Professor
Pontificia Universidad Católica de Valparaíso
Comments and Suggestions for Authors
Reviewer 1
- I do not see much improvements in the current revised manuscript, except that the author has changed some words.
In the previous review, we sent the responses to the 7 points indicated, as answered in the letter.
In this review, we incorporate the new suggestions, which are highlighted in red.
- In this study there is no extraction, measurement and evaluation of any volatile compounds, therefore both new titles are incorrect. You cannot give those tiles and assume that there was a compound. The experiments of sections 2.3 and 2.4 are simply the same (dual cultures), just used different methods.
We made the requested change. However, the terms used in the previous version are described in works where they do not use extraction methods, such as: Boiu-Sicuia OA, Toma RC, Diguță CF, Matei F, Cornea CP. In Vitro Evaluation of Some Endophytic Bacillus to Potentially Inhibit Grape and Grapevine Fungal Pathogens. Plants (Basel). 2023 Jul 5;12(13):2553. doi: 10.3390/plants12132553. PMID: 37447114; PMCID: PMC10347234.
- There is no improvement in discussion as still lots of part of results have been repeated there.
Part of the results were eliminated from the discussion and transferred to the results section on page 15, next to Table 1. Likewise, we enriched the discussion by comparing the results obtained, considering:
- Evaluation of biocontrol at different temperatures
- The importance of using more than one pathogen isolate in biocontrol assays
- The use of more than one antagonism assessment method
- Contrast of the results obtained with Pseudomonas and Rhodococcus and comparison with other species
